# WHERE OFF-POLICY DEEP REINFORCEMENT LEARNING FAILS

## ABSTRACT

This work examines batch reinforcement learning–the task of maximally exploiting a given batch of off-policy data, without further data collection. We demonstrate that due to errors introduced by extrapolation, standard off-policy deep reinforcement learning algorithms, such as DQN and DDPG, are only capable of learning with data correlated to their current policy, making them ineffective for most off-policy applications. We introduce a novel class of off-policy algorithms, batch-constrained reinforcement learning, which restricts the action space to force the agent towards behaving on-policy with respect to a subset of the given data. We extend this notion to deep reinforcement learning, and to the best of our knowledge, present the first continuous control deep reinforcement learning algorithm which can learn effectively from uncorrelated off-policy data.

## 1 INTRODUCTION

Learning from only a batch of data without requiring further interactions with the environment is a crucial requirement for scaling reinforcement learning to tasks where the data collection procedure is costly, risky, or time-consuming. Off-policy batch reinforcement learning has important implications in many areas, such as robotics, imitation learning and exploration. In practical applications, it is often preferable for data collection to be performed by some secondary controlled process, such as a human operator or a carefully monitored program. If assumptions on the quality of the behavioral policy can be made, imitation learning can be used to produce good policies. However, imitation learning algorithms, such as behavioral cloning, are known to fail when exposed to noisy or suboptimal trajectories (Ross et al., 2011), and more sophisticated approaches are generally unable to surpass the performance of the demonstrator without further interactions with the environment (Ziebart et al., 2008; Ho & Ermon, 2016; Hester et al., 2017). In most deep reinforcement learning algorithms, a single policy, or variations of a single policy, are used for both exploration and exploitation. This paradigm creates a trade-off between future learning and current performance, and limits the agent to suboptimal policies, unless careful annealing of the exploratory behavior is performed. In contrast, batch reinforcement learning offers a mechanism for learning from a decoupled exploratory process without suffering from degradation in current performance.

Most modern off-policy deep reinforcement learning algorithms fall into the category of *growing batch learning* (Lange et al., 2012), in which data is collected and stored into an experience replay dataset (Lin, 1992), which is used to train the agent before further data collection occurs. However, we find that these off-policy algorithms are generally unsuccessful, unless the available dataset is correlated to the current policy. We demonstrate that this inability to truly learn off-policy is largely due to *extrapolation error*, a phenomenon in which unseen state-action pairs in the learning target are erroneously estimated to have unrealistic values, which may make them more attractive than observed actions. In other words, the uncertainty of unseen state-action pairs and generalization from function approximation, can cause poor value estimates and suboptimal action selection.

To overcome extrapolation error in off-policy learning, we introduce batch-constrained reinforcement learning, in which agents are trained to maximize reward *by only selecting previously seen actions*. We prove that this paradigm induces desirable properties in the discrete MDP setting, such as converging to the optimal policy without needing to experience all possible transitions. We generalize this notion to continuous control with neural networks by training a state-conditioned generative model to produce only previously seen actions. This generative model is combined with

a network which aims to optimally perturb the generated actions in a small range, resulting in the agent using only actions similar to those previously seen.

We test our algorithm on several MuJoCo (Todorov et al., 2012) environments, where extrapolation error is particularly problematic due to the high dimensional continuous action space, which makes it impossible to sample the action space exhaustively. Unlike any previous continuous control reinforcement learning algorithm, our approach is able to learn successfully in a variety of batch reinforcement learning settings. Our algorithm offers a unified view on imitation and off-policy learning, and is capable of learning from purely expert demonstrations, as well as from finite batches of suboptimal data. To ensure reproducibility, we provide precise experimental and implementation details and our code will be made available.

## 2 BACKGROUND

In reinforcement learning, an agent interacts with its environment, typically assumed to be a Markov decision process (MDP) $(\mathcal{S}, \mathcal{A}, p, r, \gamma)$, with state space $\mathcal{S}$, action space $\mathcal{A}$ transition dynamics $p(s'|s, a)$, for each $s', s \in \mathcal{S}$ and $a \in \mathcal{A}$. At each discrete time step, the agent receives a reward $r(s, a) \in \mathbb{R}$ for performing action $a$ in state $s$. The goal of the agent is to maximize the expectation of the sum of discounted rewards, known as the return $R_t = \sum_{i=t+1}^{\infty} \gamma^i r(s_i, a_i)$, which weighs future rewards with respect to the discount factor $\gamma \in [0, 1)$.

The agent selects actions with respect to a policy $\pi : \mathcal{S} \rightarrow \mathcal{A}$, which has a corresponding value function $Q^\pi(s, a) = \mathbb{E}_\pi[R_t|s, a]$, estimating the expected return when following the policy $\pi$ after taking action $a$ in state $s$. Given $Q^\pi$, a new policy $\pi'$ of equal or better performance can be derived by greedy maximization $\pi' = \operatorname{argmax}_a Q^\pi(s, a)$ (Sutton & Barto, 1998). For a given policy $\pi$, the value function can be estimated using sampled versions of the Bellman operator $\mathcal{T}^\pi$:

$$\mathcal{T}^\pi Q(s, a) = \mathbb{E}_{s'}[r + \gamma Q(s', \pi(s'))]. \tag{1}$$

The Bellman operator is a contraction for $\gamma \in [0, 1)$ with unique fixed point $Q^\pi(s, a)$ (Bertsekas & Tsitsiklis, 1996). $Q^*(s, a) = \max_\pi Q^\pi(s, a)$ is known as the optimal value function, which has a corresponding optimal policy obtained through greedy action choices. For large or continuous state and action spaces, the value can be approximated with neural networks, e.g. using the DQN algorithm (Mnih et al., 2015). In DQN, the value function $Q_\theta$ is updated using the target:

$$r + \gamma Q_{\theta'}(s', \pi(s')), \quad \pi(s') = \operatorname*{argmax}_a Q_{\theta'}(s', a), \tag{2}$$

Q-learning is an *off-policy* algorithm (Sutton & Barto, 1998), meaning the target can be computed without consideration of how the experience was generated. As a result, off-policy reinforcement learning algorithms are able to learn from data collected by any behavioral policy. Typically, the loss is minimized over mini-batches of tuples of the agent's past data, $(s, a, r, s') \in \mathcal{B}$, sampled from an experience replay dataset $\mathcal{B}$ (Lin, 1992). For shorthand, we often write $s \in \mathcal{B}$ if there exists a transition tuple containing $s$ in the batch $\mathcal{B}$, and similarly for $(s, a) \in \mathcal{B}$. To further stabilize learning, a target network with frozen parameters $Q_{\theta'}$, is used in the learning target. The parameters of the target network $\theta'$ are updated to the current network parameters $\theta$ after a fixed number of time steps, or by averaging $\theta' \leftarrow \tau\theta + (1 - \tau)\theta'$ for some small $\tau$ (Lillicrap et al., 2015).

In a continuous action space, the analytic maximum of Equation 2 is intractable. In this case, actor-critic methods are commonly used, where action selection is performed through a separate policy network $\pi_\phi$, known as the actor, and updated with respect to a value estimate, known as the critic (Sutton & Barto, 1998; Konda & Tsitsiklis, 2003). This policy can be updated following the deterministic policy gradient theorem (Silver et al., 2014):

$$\phi = \operatorname*{argmax}_{\tilde{\phi}} \mathbb{E}_{s \in \mathcal{B}}[Q_\theta(s, \pi_{\tilde{\phi}}(s))], \tag{3}$$

which corresponds to learning an approximation to the maximum of $Q_\theta$, by propagating the gradient through both $\pi_\phi$ and $Q_\theta$. When combined with off-policy deep Q-learning to learn $Q_\theta$, this algorithm is referred to as Deep Deterministic Policy Gradients (DDPG) (Lillicrap et al., 2015).

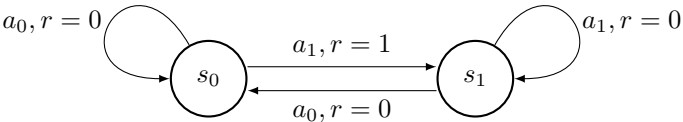

Figure 1: Toy MDP with two states $s_0$ and $s_1$, and two actions $a_0$ and $a_1$. Agent receives reward of 1 for selecting $a_1$ at $s_0$ and 0 otherwise.

## 3 EXTRAPOLATION ERROR

Off-policy algorithms commonly assume infinite state-action visitation for convergence or optimality guarantees (Watkins, 1989; Precup et al., 2001), which is generally impossible to satisfy in any practical problem. In this section, we analyze how breaking this assumption can introduce large amounts of error into the value function by a phenomenon we call *extrapolation error*. We show how this error can lead to degenerate policies for even provably convergent batch algorithms, such as kernel-based reinforcement learning (Ormoneit & Sen, 2002). Furthermore, we find that state of the art deep reinforcement learning algorithms, such as DDPG (Lillicrap et al., 2015) and DQN (Mnih et al., 2015), are generally incapable of learning from data uncorrelated to the current policy. Consequently, current deep reinforcement learning algorithms are unsatisfactory solutions for the batch setting, where no assumptions can be made on the data collection process.

In Q-learning, an off-policy target $r + \gamma Q(s', a')$ is used to train the current value estimate, where $a'$ is selected by a policy $\pi(s')$. However, if the state-action pair $(s', a')$ is not contained in the training set, the value $Q(s', a')$ can be unpredictable due to extrapolation from other state-action pairs by function approximation. This means $Q(s', a')$ can take on implausible values due to the lack of data near the state-action pair. Informally, extrapolation error comes from training the value estimate with state-action pairs that are not present in the training data. When combined with maximization in reinforcement learning algorithms, extrapolation error provides a source of noise that can induce a persistent overestimation bias (Thrun & Schwartz, 1993; Van Hasselt et al., 2016; Fujimoto et al., 2018). In an off-policy or batch setting, this extrapolation error may never be corrected, due to the inability to collect new data, resulting in disastrous value estimation and degenerate policies.

### 3.1 A SIMPLE EXAMPLE

This problem of extrapolation persists in traditional batch reinforcement learning algorithms, such as kernel-based reinforcement learning (KBRL) (Ormoneit & Sen, 2002). For a given batch $\mathcal{B}$ of transitions $(s, a, r, s')$, non-negative density function $\phi : \mathbb{R}^+ \to \mathbb{R}^+$, hyper-parameter $\tau \in \mathbb{R}$, and norm $|| \cdot ||$, KBRL evaluates the value of a state-action pair (s,a) as follows:

$$Q(s, a) = \sum_{(s_{\mathcal{B}}^a, a, r, s_{\mathcal{B}}') \in \mathcal{B}} \kappa_\tau^a(s, s_{\mathcal{B}}^a)[r + \gamma V(s_{\mathcal{B}}')], \tag{4}$$

$$\kappa_\tau^a(s, s_{\mathcal{B}}^a) = \frac{k_\tau(s, s_{\mathcal{B}}^a)}{\sum_{\tilde{s}_{\mathcal{B}}^a} k_\tau(s, \tilde{s}_{\mathcal{B}}^a)}, \qquad k_\tau(s, s_{\mathcal{B}}^a) = \phi\left(\frac{||s - s_{\mathcal{B}}^a||}{\tau}\right), \tag{5}$$

where states $s_{\mathcal{B}}^a \in \mathcal{S}$ corresponding to the action $a$ $(s_{\mathcal{B}}, a) \in \mathcal{B}$, and $V(s_{\mathcal{B}}') = \max_{a \text{ s.t.}(s_{\mathcal{B}}', a) \in \mathcal{B}} Q(s_{\mathcal{B}}', a)$. At each iteration, KBRL updates the estimates of $Q(s_{\mathcal{B}}, a_{\mathcal{B}})$ for all $(s_{\mathcal{B}}, a_{\mathcal{B}}) \in \mathcal{B}$ following Equation (4), then updates $V(s_{\mathcal{B}}')$ by evaluating $Q(s_{\mathcal{B}}', a)$ for all $s_{\mathcal{B}} \in \mathcal{B}$ and $a \in \mathcal{A}$.

Given access to the entire deterministic MDP, KBRL will provable converge to the optimal value, however when limited to only a subset, we find the value estimation susceptible to extrapolation. In Figure 1, we provide a deterministic two state, two action MDP in which KBRL fails to learn the optimal policy when provided with state-action pairs from the optimal policy. Given the batch $\{(s_0, a_1, r = 1, s_1), (s_1, a_0, r = 0, s_0)\}$, corresponding to the optimal behavior, and noting that there is only one example of each action, Equation (4) provides the following:

$$Q(\cdot, a_1) = 1 + \gamma V(s_1) = 1 + \gamma Q(s_1, a_0), \qquad Q(\cdot, a_0) = \gamma V(s_0) = \gamma Q(s_0, a_1). \tag{6}$$

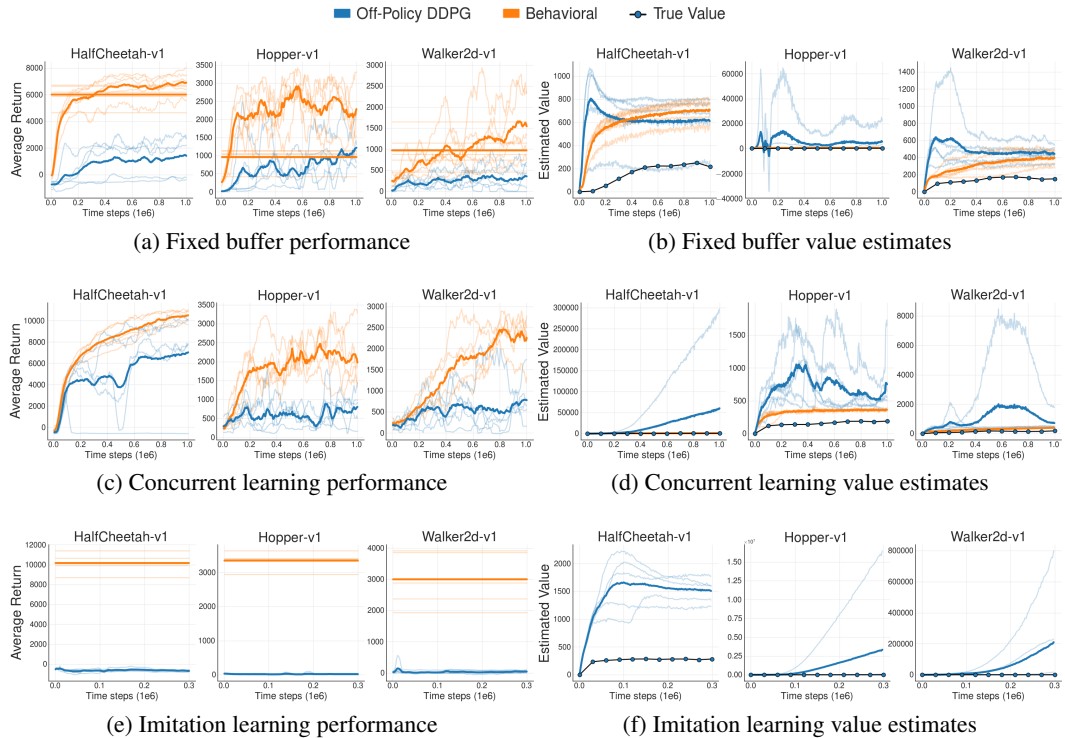

Figure 2: We examine the performance of DDPG in three off-policy settings. Each individual trial is plotted with a thin line, with the mean in bold. Straight lines represent the average return of episodes contained in the batch. An estimate of the true value of the off-policy agent, evaluated by Monte Carlo returns, is marked by a dotted line. In the fixed buffer experiment, the off-policy agent learns from a large, diverse dataset, but exhibits poor learning behavior and value estimation. In the concurrent learning setting the agent learns alongside a behavioral agent, with access to the same data, but suffers in performance. In the imitation learning setting, the agent receives data from an expert policy but is unable to learn, and exhibits highly divergent value estimates.

After sufficient iterations KBRL will converge correctly to $Q(s_0, a_1) = \frac{1}{1-\gamma^2}$, $Q(s_1, a_0) = \frac{\gamma}{1-\gamma^2}$. However, when evaluating actions, KBRL erroneously extrapolates the values of each action $Q(\cdot, a_1) = \frac{1}{1-\gamma^2}$, $Q(\cdot, a_0) = \frac{\gamma}{1-\gamma^2}$, and its behavior, $\mathrm{argmax}_a Q(s, a)$, will result in the degenerate policy of continually selecting $a_1$. KBRL fails this example by estimating the values of unseen state-action pairs. In methods where the extrapolated estimates can be included into the learning update, such fitted Q-iteration or DQN (Ernst et al., 2005; Mnih et al., 2015), this can cause an increasing sequence of value estimates.

## 3.2 EXTRAPOLATION ERROR IN DEEP REINFORCEMENT LEARNING

While Section 3.1 examined a specific counterexample, this raises a key question–*what is the role of extrapolation error in a practical setting?* In this section, we examine the behavior of a state of the art deep actor-critic algorithm, DDPG (Lillicrap et al., 2015), when learned with off-policy data. We find that extrapolation error, combined with the maximization via gradient updates in the policy updates of DDPG, results in large amounts of overestimation bias and reduced performance. To examine the bias induced from off-policy learning, we examine three different batch settings in the MuJoCo environments of OpenAI gym (Todorov et al., 2012; Brockman et al., 2016), which we use to train an off-policy DDPG agent with no interaction with the environment:

**Batch 1 (Fixed buffer).** We train a DDPG (Lillicrap et al., 2015) agent for 1 million time steps, adding large amounts of Gaussian noise, $\mathcal{N}(0, 0.5)$, for exploration, and store all experienced transitions. This collection procedure creates a dataset with a diverse set of states and actions.

**Batch 2 (Concurrent learning).** We simultaneously train the off-policy DDPG agent with the behavioral DDPG agent, for 1 million time steps. The behavioral policy performs data collection

with $\mathcal{N}(0, 0.1)$ Gaussian noise added for exploration. Each transition experienced by the behavioral policy is stored in a buffer replay which both agents learn from. Batch 2 differs from Batch 1 as the agent learns as the buffer replay grows.

**Batch 3 (Imitation).** A trained DDPG agent acts as an expert, and is used to collect a dataset of 1 million transitions, without added noise. For each experiment we graph the performance of the agents as they train (left), as well as their value estimates (right), in Figure 2. Straight lines represent the average return of episodes contained in the batch. For better comparison, we additionally graph the learning performance of the behavioral agent for the fixed buffer experiment. Naïvely, we might expect the off-policy agent trained with Batch 1 to perform well, given the large amounts of exploration, and size of the buffer, however, we find that the agent performs far worse than the behavioral agent. Furthermore, we find the value function overestimates the true value.

In experiments with Batch 2, both agents sees the same buffer, we might expect the performance to be very similar. However, due to small differences in the initial policies, the impact of extrapolation error on the off-policy agent is observed. There is a notable difference in performance, and the value estimates of the off-policy agent is consistently higher than the behavioral value estimates.

Results of the agent trained with Batch 3 demonstrates the failure of off-policy deep reinforcement learning algorithms when the state-action coverage is limited, as exploration is hampered due to the unchanging expert policy. In this experiment the off-policy agent fails to learn any meaningful behavior and the value function demonstrates divergent behavior.

These experiments show extrapolation error can be highly detrimental to learning off-policy in a batch reinforcement learning setting. While the continuous state space and multi-dimensional action spaces in MuJoCo environments are contributing factors to extrapolation error, the scale of these tasks is small compared to real world settings. However, even if a sufficient amount of data collection occurs, the concerns of catastrophic forgetting (McCloskey & Cohen, 1989; Goodfellow et al., 2013) may still result in extrapolation error. Consequently, off-policy reinforcement learning algorithms used in the real-world will require practical guarantees without infinite data.

## 4 OFF-POLICY REINFORCEMENT LEARNING WITHOUT INFINITE DATA

Extrapolation error is introduced when we evaluate a learning target with an *unseen* state-action pair, during off-policy learning without infinite data. To address this concern, in this section we propose a learning agent which aims to act optimally while selecting actions from corresponding *seen* state-action pairs. In other words, for a given batch, we aim to learn the optimal agent which is on-policy with respect to a subset of the provided data. We begin by deriving the batch-constrained policy iteration algorithm in a discrete MDP setting in Section 4.1 and then extend the algorithm to a deep reinforcement learning setting in Section 4.2.

### 4.1 BATCH-CONSTRAINED REINFORCEMENT LEARNING FOR DISCRETE MDPS

While extrapolation error is induced by function approximation, this section aims to introduce theoretical concepts and explain why a constrained action-space approach is a desirable strategy, even in a discrete MDP setting without function approximation. To begin formalizing the notion of only learning with seen data, we define a batch $\mathcal{B}$ as a set of tuples $(s, a, r, s')$, where we further assume if $(s, a) \in \mathcal{B}$ then $s' \in \mathcal{B}$ if $p(s'|s, a) > 0$ unless $s'$ is a terminal state. For a given batch $\mathcal{B}$, we define the set of batch-constrained policies $\Pi_\mathcal{B}$, where $\pi \in \Pi_\mathcal{B} : \mathcal{S} \to \mathcal{A}$, maps states to actions with the condition that if $s \in \mathcal{B}$ then the policy must select previously seen actions such that $(s, \pi(s)) \in \mathcal{B}$.

In a practical reinforcement learning setup, the policy can only be trained with previously collected state-action pairs, thus we define the batch Bellman operator $\mathcal{T}_\mathcal{B}^\pi$, which will produce $\infty$ if a state-action pair is not contained in the batch:

$$\mathcal{T}_\mathcal{B}^\pi Q(s, a) = \begin{cases} \mathbb{E}_{s'}[r + \gamma Q(s', \pi(s'))] & \text{if } (s, a) \in \mathcal{B} \\ \infty & \text{otherwise.} \end{cases} \tag{7}$$

The batch Bellman operator serves as a representation for extrapolation error, as it will produce $\infty$ in regions that are uncertain due to unseen data points. For a given batch $\mathcal{B}$ and policy $\pi$, we can use the batch Bellman operator to compute the value of a batch-constrained policy $\pi \in \Pi_\mathcal{B}$.

**Lemma 1 (Batch Bellman Policy Evaluation).** *For a given batch $\mathcal{B}$, MDP $M$, and batch-constrained policy $\pi \in \Pi_{\mathcal{B}}$, the batch Bellman operator $\mathcal{T}_{\mathcal{B}}^{\pi}$ repeatedly applied to an initial $Q_0$ : $\mathcal{S} \times \mathcal{A} \to \mathbb{R}$, where $Q_{k+1} = \mathcal{T}_{\mathcal{B}}^{\pi} Q_k$ converges to a fixed point $Q_{\mathcal{B}}^{\pi}(s,a)$ where $Q_{\mathcal{B}}^{\pi}(s,a) = Q^{\pi}(s,a)$ for $(s,a) \in \mathcal{B}$, as $k \to \infty$.*

The proof follows naturally by constructing a new MDP with transitions defined by the batch. Lemma 1 states if $\pi \in \Pi_{\mathcal{B}}$, the resulting value function $Q_{\mathcal{B}}^{\pi}(s,a)$ will be accurate, i.e. $Q_{\mathcal{B}}^{\pi}(s,a) = Q^{\pi}(s,a)$, at any state-action pair $(s,a)$ contained in the batch $\mathcal{B}$. This Lemma leads us to a key result concerning batch-constrained policies, which states only batch-constrained policies can produce accurate value functions with respect to the batch Bellman operator.

**Theorem 1 (Batch Bellman Necessary and Sufficient Condition).** *Let $Q_{\mathcal{B}}^{\pi}(s,a)$ be the fixed point of the batch Bellman operator $\mathcal{T}_{\mathcal{B}}^{\pi}$, then $Q_{\mathcal{B}}^{\pi}(s,a) = Q^{\pi}(s,a)$ for all $(s,a) \in \mathcal{B}$ if and only if $\pi \in \Pi_{\mathcal{B}}$.*

The batch Bellman operator overestimates the value for unseen states, analogous to the worst-case of extrapolation error in a function approximation setting. Theorem 1 implies with access to only a subset of state-action pairs in the MDP, the value function $Q_{\mathcal{B}}^{\pi}$ learned with the batch Bellman operator will be inaccurate unless $\pi$ is batch-constrained. Given the batch Bellman operator is a representation of an extreme case of extrapolation error, Theorem 1 suggests batch-constrained policies are a potential solution to limiting incorrect extrapolation in the standard setting.

Next, we demonstrate batch-constrained policies can be used to learn the optimal policy. We first generalize the policy improvement theorem to a batch-constrained setting. Given the value function of any batch-constrained policy, a policy of equal or higher value can be computed by greedily selecting actions $a$ of highest value that satisfy the batch constraint $(s,a) \in \mathcal{B}$.

**Lemma 2 (Batch-Constrained Policy Improvement).** *Let $\pi \in \Pi_{\mathcal{B}}$ be any deterministic batch-constrained policy and $\pi'(s) = \mathrm{argmax}_{a \text{ s.t.} (s,a) \in \mathcal{B}} Q^{\pi}(s,a)$ , then for all $s \in \mathcal{B}$, $Q^{\pi'}(s, \pi'(s)) \geq Q^{\pi}(s, \pi(s))$.*

The previous Lemma, along with the standard policy evaluation result (Sutton & Barto, 1998) can be brought together to provide a policy iteration result. By repeatedly evaluating an initial batch-constrained policy, and greedily maximizing with respect to value function and batch constraints, the policy converges to the optimal batch-constrained policy.

**Theorem 2 (Batch-Constrained Policy Iteration).** *Given a batch $\mathcal{B}$ and MDP $M$, then the repeated application of policy evaluation and batch-constrained policy improvement converges to a policy $\pi_{\mathcal{B}}^{*}$, such that $Q^{\pi_{\mathcal{B}}^{*}}(s, \pi_{\mathcal{B}}^{*}(s)) \geq Q^{\pi_{\mathcal{B}}}(s, \pi_{\mathcal{B}}(s))$ for all $\pi_{\mathcal{B}} \in \Pi_{\mathcal{B}}$ and $s \in \mathcal{B}$.*

Optimality is a direct result from Theorem 1, namely, if the batch contains all possible state-action pairs from an optimal policy $\pi^{*}$, batch-constrained policy iteration converges to the optimal policy.

**Corollary 1 (Batch-Constrained Optimality).** *Given a batch $\mathcal{B}$, and MDP $M$ if for all $s \in \mathcal{S}$, $(s, a^{*}, r, s') \in B$ where $a^{*} = \mathrm{argmax}_{a} Q^{*}(s,a)$ and $Q^{*}(s,a) = \max_{\pi} Q^{\pi}(s,a)$, then batch-constrained policy iteration converges to a policy $\pi^{*} = \mathrm{argmax}_{a} Q^{*}(s,a)$ for all $s \in \mathcal{S}$.*

This result can be extended to show the direct connection from batch-constrained policy iteration and imitation learning. Given an optimal trajectory, batch-constrained policy iteration converges to a policy which imitates the optimal trajectory over the state space it covers. This result demonstrates that batch-constrained policy iteration is able to learn meaningful policies in certain regions of the state space, without access to the entire MDP.

**Corollary 2 (Batch-Constrained Imitation).** *Given a deterministic MDP, and trajectory $T = (s_0, a_0^{*}, r_0, ..., s_{i+1}, a_{i+1}^{*}, r_{i+1}, ...)$ where $a_i^{*} = \mathrm{argmax}_{a} Q^{*}(s_i, a)$ and $Q^{*}(s,a) = \max_{\pi} Q^{\pi}(s,a)$, then batch-constrained policy iteration converges to $\pi^{*}(s_i) = \mathrm{argmax}_{a} Q^{*}(s_i, a)$ for all $s_i \in T$.*

## 4.2 BETWEEN IMITATION AND REINFORCEMENT LEARNING FOR CONTINUOUS CONTROL

In this section we use the theoretical concepts introduced in the previous section as inspiration for a deep reinforcement learning algorithm in the continuous control setting, which minimizes extrapolation error through an approximation to the batch-constraint. By nature, moving away from the discrete MDP setting brings a different set of challenges. For a given continuous state $s \in \mathcal{B}$, it is

unlikely that there is more than one corresponding action $a$ such that $(s, a) \in \mathcal{B}$. This is problematic as a batch-constrained policy will learn to simply imitate previously taken actions, which may be undesirable if the batch contains any suboptimal actions. To address this, we define a similarity metric by modelling a learned distribution over actions contained in the batch. This enables our deep batch reinforcement learning algorithm, Batch-Constrained deep Q-learning (BCQ), to approximate the batch-constraint by selecting actions which are similar to previously seen actions.

The batch-constraint was introduced as a tool to reduce extrapolation error by selecting only actions from previously seen state-action pairs. We define a similarity metric by making the assumption that the similarity between $(s, a)$ and the state-action pairs in the batch $\mathcal{B}$ can be modelled using a learned state-conditioned marginal likelihood $P_{\mathcal{B}}(a|s)$. In this case, it follows that that the policy maximizing $\mathbb{E}_{a \sim \pi}[P_{\mathcal{B}}(a|s)]$ would minimize the error induced by extrapolation from distant, or unseen, state-action pairs, by only selecting the most likely actions in the batch, with respect to a given state. Given the difficulty of directly maximizing $P_{\mathcal{B}}(a|s)$ due to its intractable nature, we instead train a parametric generative model of the batch $G_{\omega}(s)$, which we can sample actions from as a reasonable approximation to $\operatorname{argmax}_{\pi} \mathbb{E}_{a \sim \pi}[P_{\mathcal{B}}(a|s)]$.

For our generative model we use a conditional variational auto-encoder (VAE) (Kingma & Welling, 2013; Sohn et al., 2015), which models the distribution by transforming an underlying latent space[1]. The encoder-decoder generative model $G_{\omega} = \{E_{\omega_1}, D_{\omega_2}\}$ is trained to maximize the marginal likelihood of the batch $\mathcal{B}$ by optimizing the variational lowerbound:

$$\omega = \operatorname*{argmin}_{\tilde{\omega}} \sum_{(s,a) \in \mathcal{B}} (D_{\tilde{\omega}_2}(s, z) - a)^2 + D_{\mathrm{KL}}(\mathcal{N}(\mu, \sigma) || \mathcal{N}(0, 1)), \tag{8}$$

where $E_{\tilde{\omega}_1}(s, a) = \mu, \sigma$ and $z \sim \mathcal{N}(\mu, \sigma)$. To reduce extrapolation error in the value estimate $Q_{\theta}$, actions which are similar to previously seen actions need to be selected in the learning target $r + \gamma Q_{\theta'}(s', \pi(s'))$. The generative model $G_{\omega}$ provides this mechanism, as for a given state, actions similar to the actions in the batch can sampled with high probability.

The generative model $G_{\omega}$, alongside the value function $Q_{\theta}$, can be used as a policy by sampling $n$ actions from $G_{\omega}$ and selecting the highest valued action according to the value estimate $Q_{\theta}$. To increase the diversity of seen actions, we introduce a perturbation model $\xi_{\phi}(s, a, \Phi)$, which outputs an adjustment to an action $a$ in the range $[-\Phi, \Phi]$. This enables access to actions in a constrained region, without having to sample from the generative model a prohibitive number of times. This results in a policy $\pi$ which is defined by the hyper-parameters $n$ and $\Phi$, the value function $Q_{\theta}$ and the perturbation model $\xi_{\phi}$:

$$\pi(s) = \operatorname*{argmax}_{a_i} Q_{\theta}(s, a_i + \xi_{\phi}(s, a_i, \Phi)), \quad \{a_i \sim G_{\omega}(s)\}_{i=1}^n. \tag{9}$$

The choice of $n$ and $\Phi$ creates a trade-off between an imitation learning and reinforcement learning algorithm. If $\Phi = 0$, and the number of sampled actions $n = 1$, then the policy resembles behavioral cloning and as $\Phi \to a_{\max} - a_{\min}$ and $n \to \infty$, then $\pi = \operatorname{argmax}_a Q_{\theta}(s', a)$, approaching Q-learning as the policy becomes closer to greedy maximization of the value function.

The perturbation model $\xi_{\phi}$ can be trained to maximize $Q_{\theta}(s, a)$ through the deterministic policy gradient algorithm (Silver et al., 2014):

$$\phi = \operatorname*{argmax}_{\tilde{\phi}} \sum_{(s,a) \in \mathcal{B}} Q_{\theta}(s, a + \xi_{\tilde{\phi}}(s, a, \Phi)), \quad a \sim G_{\omega}(s). \tag{10}$$

As $\pi$ includes a random sampling process, the variance of the value estimate update can be reduced by including a value network $V_{\psi}(s)$, which changes the learning target of $Q_{\theta}$ to $r + \gamma V_{\psi'}(s')$. This value network can be updated with learning target:

$$\max_{a_i} Q_{\theta}(s, a_i + \xi_{\phi}(s, a_i, \Phi))), \quad a_i \sim G_{\omega}(s). \tag{11}$$

This forms Batch-Constrained deep Q-learning (BCQ), which maintains four parametrized networks, a perturbation model $\xi_{\phi}(s, a)$, a critic $Q_{\theta}(s, a)$, a value network $V_{\psi}(s)$, and a generative model $G_{\omega}(s)$. We summarize BCQ in Algorithm 1. While BCQ is an approximation to batch-constrained reinforcement learning, in the following section we demonstrate that this approximation results in stable value learning and a strong performance in the batch setting. Furthermore, we find that only a single setting of $n$ and $\Phi$ is necessary for a wide range of tasks.

---

[1] For an introduction to VAEs, see Supplementary Material E.

---

**Algorithm 1** BCQ

---

**Input:** Batch $\mathcal{B}$, horizon $T$, target network update rate $\tau$, mini-batch size $N$, max perturbation $\Phi$
Initialize critic network $Q_\theta$, value network $V_\psi$, perturbation network $\xi_\phi$, and VAE $G_\omega = \{E_{\omega_1}, D_{\omega_2}\}$, with random parameters $\theta, \psi, \phi, \omega$, and target network $V_{\psi'}$ with $\psi' \leftarrow \psi$
**for** $t = 1$ **to** $T$ **do**
    Sample mini-batch of $N$ transitions $(s, a, r, s')$ from $\mathcal{B}$
    Sample VAE: $\mu, \sigma = E_{\omega_1}(s, a), \quad \tilde{a} = D_{\omega_2}(s, z), \quad z \sim \mathcal{N}(\mu, \sigma)$
    Update VAE: $\omega \leftarrow \operatorname{argmin}_\omega \sum (a - \tilde{a})^2 + D_{\mathrm{KL}}(\mathcal{N}(\mu, \sigma) \| \mathcal{N}(0, 1))$
    Update critic: $\theta \leftarrow \operatorname{argmin}_\theta \sum (r + \gamma V_{\psi'}(s') - Q_\theta(s, a))^2$
    Update perturbation model: $\phi \leftarrow \operatorname{argmax}_\phi \sum Q(s, a + \xi_\phi(s, a, \Phi)), \quad a \sim G_\omega(s)$
    Sample $n$ actions: $\{a_i \sim G_\omega(s')\}_{i=1,...,n}$
    Perturb each action: $\{a_i = a_i + \xi_\phi(s', a_i, \Phi)\}_{i=1,...,n}$
    Update value: $\psi \leftarrow \operatorname{argmin}_\psi \sum (\max_{a_i} Q_\theta(s, a_i) - V_\psi(s))^2$
    Update target network: $\psi' \leftarrow \tau\psi + (1 - \tau)\psi'$
**end for**

---

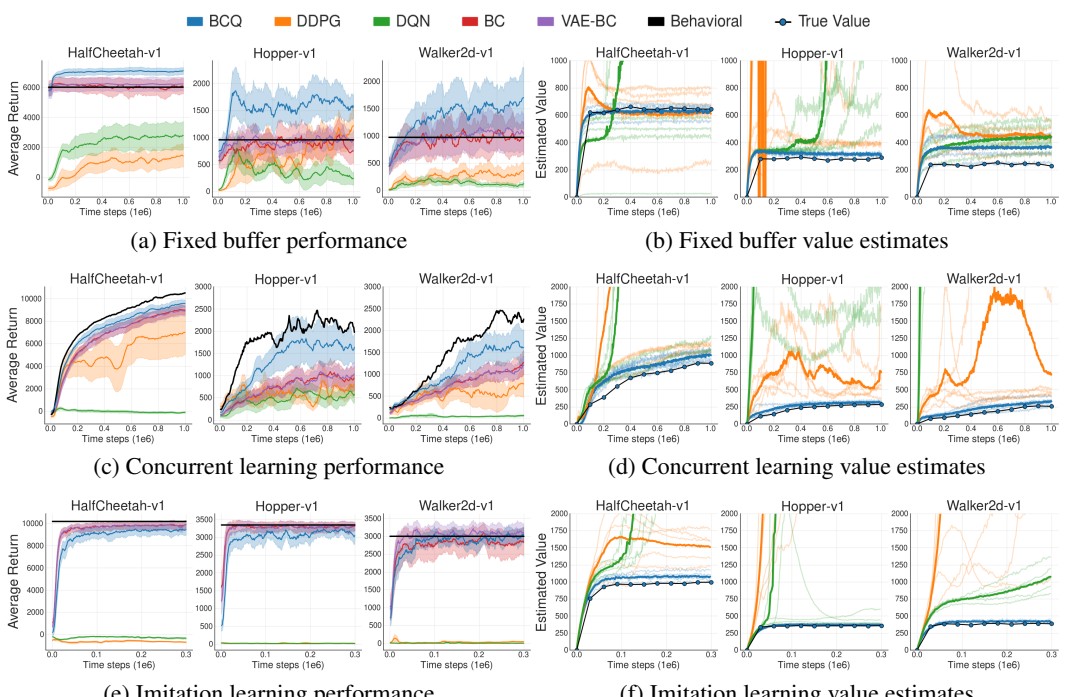

(a) Fixed buffer performance          (b) Fixed buffer value estimates

(c) Concurrent learning performance      (d) Concurrent learning value estimates

(e) Imitation learning performance      (f) Imitation learning value estimates

Figure 3: We evaluate BCQ and several baselines following the experiments from Section 3.2. The shaded area represents half a standard deviation. Value estimates include a plot of each trial, with the mean in bold. Straight lines represent the average return of episodes contained in the batch. An estimate of the true value of BCQ, evaluated by Monte Carlo returns, is marked by a dotted line. Unlike any other algorithm, BCQ matches or outperforms the performance of the behavioral policy in all three tasks, while exhibiting a stable value function.

## 5 EXPERIMENTS

To evaluate the effectiveness of Batch-Constrained deep Q-learning (BCQ) in a high-dimensional setting, we focus on MuJoCo environments in OpenAI gym (Todorov et al., 2012; Brockman et al., 2016). For reproducibility, we make no modifications to the original environments or reward functions. We compare our method with DDPG (Lillicrap et al., 2015), DQN (Mnih et al., 2015) using an independently discretized action space with 10 bins per dimension, a feed-forward behavioral cloning method (BC), and a variant with a VAE (VAE-BC), mimicking $G_\omega(s)$ used by BCQ. For BCQ, we use $\Phi = 0.05$, sample $n = 10$ actions from the VAE $G_\omega$ and set the latent space to have twice the number of dimensions as the action space. Exact network, hyper-parameter and training details of each method, along with experimental details are provided in the Supplementary Material.

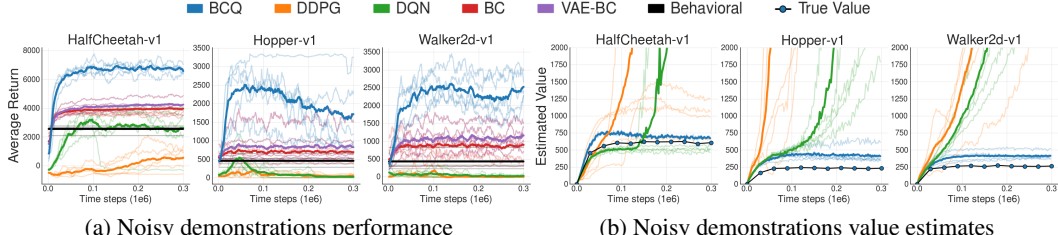

(a) Noisy demonstrations performance    (b) Noisy demonstrations value estimates

Figure 4: We examine the effectiveness of BCQ when learning from a highly noisy demonstrator. 100k transitions are provided by an expert policy with $0.3$ probability of a random action and $\mathcal{N}(0, 0.3)$ added to remaining actions. Average returns include a plot of each trial, with the mean in bold. BCQ greatly outperforms behavioral cloning algorithms, as well as the behavioral policy, demonstrating robustness to suboptimal data.

We evaluate each method following the three experiments defined in Section 3.2: *fixed buffer*, learning from the final experience replay of a trained DDPG agent, *concurrent learning*, learning simultaneously with the behavioral DDPG policy, and *imitation*, learning from a dataset collected by an expert. The results of these experiments, along with the estimated values of BCQ, DDPG and DQN, and the true value of BCQ are displayed in Figure 3. Our approach, BCQ, is the only algorithm which succeeds at all three tasks, matching or outperforming the behavioral policy in each instance, and outperforming all other agents, besides in imitation learning where behavioral cloning unsurprisingly performs the best. These result demonstrate our approach is capable of both performing imitation learning and off-policy learning with fixed hyper-parameters. Furthermore, unlike DDPG and DQN, BCQ exhibits a highly stable value function in the presence of off-policy samples, suggesting extrapolation error has been successfully mitigated.

Given the action space of DQN is discretized independently, it is foreseeable that it would perform poorly on these environments with multi-dimensional actions. However, this discretization reduces the action space to a more manageable scale, allowing for greater coverage of the action space, and potentially reducing the impact of extrapolation error. Unfortunately, it is clear that this hypothesis is false, and the greedy maximization of DQN produces a highly overestimated value function. Alongside extrapolation error, a likely cause is the error introduced by the independent discretization. These results suggest that naïvely reducing the size of the action space of the environment is an inadequate solution to extrapolation error.

To study the robustness of BCQ to imperfect transitions and multi-modal data, we examine its performance with a batch of 100k transitions collected by an expert policy, with two sources of noise. The behavioral policy selects actions randomly with probability $0.3$ and with high exploratory noise $\mathcal{N}(0, 0.3)$ added to the remaining actions. We report the results in Figure 4. We find that both deep reinforcement learning and imitation learning algorithms preform poorly at this task. BCQ, however, is able to strongly outperform the performance of the noisy demonstrator, disentangling poor and expert actions. Furthermore, compared to current deep reinforcement learning algorithms, such as the DDPG demonstrator in Figure 3.c, BCQ attains a high performance in remarkably few iterations. As DDPG is known to be data-efficient compared to other state of the art algorithms (Henderson et al., 2017), this suggests our approach effectively leverages expert transitions.

## 6 RELATED WORK

**Batch Reinforcement Learning.** While batch reinforcement learning algorithms have been shown to be convergent with non-parametric function approximators such as averagers (Gordon, 1995) and kernel methods (Ormoneit & Sen, 2002), they make no guarantees on the quality of the policy without infinite data. Other batch algorithms, such as fitted Q-iteration, have used other function approximators, including decision trees (Ernst et al., 2005) and neural networks (Riedmiller, 2005), but come without convergence guarantees. Decoupling exploration and exploitation has been previously attempted by achieving sufficient diversity in data in an exploratory phase (Colas et al., 2018). Due to fewer assumptions on the data collection, off-policy algorithms which rely on importance sampling (Precup et al., 2001; Munos et al., 2016) may not be applicable in a batch setting, and scale poorly to multi-dimensional action spaces. Reinforcement learning with the experience replay (Lin, 1992) can be considered a form of batch reinforcement learning, and is a standard tool for off-

policy deep reinforcement learning algorithms (Mnih et al., 2015). It has been observed that large experience replay (Lin, 1992) can be detrimental to performance (de Bruin et al., 2015; Zhang & Sutton, 2017) and the diversity of states in the buffer is an important factor for performance (de Bruin et al., 2016). Isele & Cosgun (2018) observed the performance of agent was strongest when the distribution of data in the replay buffer matches the test distribution. These results defend the notion that extrapolation error is an important factor in the performance off-policy reinforcement learning.

**Imitation Learning.** Imitation learning and its variants are well studied problems (Schaal, 1999; Argall et al., 2009). In recent years, combining imitation with reinforcement learning, via learning from demonstrations (Kim et al., 2013; Piot et al., 2014; Chemali & Lazaric, 2015) has grown in popularity with extensions to deep reinforcement learning (Hester et al., 2017; Večerík et al., 2017). While effective, these methods require either explicit labeling of expert data, or further on-policy data collection to surpass the performance of the demonstrator. Research in imitation and inverse reinforcement learning, with robustness to noise is an emerging area (Evans, 2016; Nair et al., 2017), but require expert data. Gao et al. (2018) introduced an imitation learning algorithm which learned from highly imperfect demonstrations, by favoring seen actions, but is limited to discrete actions.

**Uncertainty in Reinforcement Learning.** Uncertainty estimates in deep reinforcement learning have generally been used to encourage exploration (Dearden et al., 1998; Strehl & Littman, 2008; O'Donoghue et al., 2018; Azizzadenesheli et al., 2018). Other methods have examined approximating the Bayesian posterior of the value function (Osband et al., 2016; 2018; Touati et al., 2018), again using the variance to encourage exploration to unseen regions of the state space. It has been previously noted that non-linear function approximation itself induces an implicit exploration without an explicit exploration policy (Dauparas et al., 2018). Our results suggest that this exploration may be the result of an extrapolation error due to unseen state-action pairs. In model-based reinforcement learning, uncertainty has been used for exploration, but also for the opposite effect–to push the policy towards regions of certainty in the model. This is used to combat the well-known problems with compounding model errors, and is present in policy search methods (Deisenroth & Rasmussen, 2011; Gal et al., 2016; Higuera et al., 2018), combined with trajectory optimization (Chua et al., 2018) or value-based methods (Buckman et al., 2018). Our work connects to policy methods with conservative updates (Kakade & Langford, 2002), such as trust region (Schulman et al., 2015; Achiam et al., 2017) and information-theoretic methods (Peters & Mülling, 2010; Van Hoof et al., 2017) which aim to keep an updated policy similar to the previous policy. These methods avoid explicit uncertainty estimates, and rather force policy updates into a constrained range before collecting new data, limiting errors introduced by large changes in the policy. Similarly, our approach can be thought of an off-policy variant, where the policy aims to be kept close, in output space, to any combination of the previous policies which performed the data collection.

## 7 CONCLUSION

In this work we demonstrate a critical problem in off-policy reinforcement learning with finite data and function approximation, where the value target introduces error by including an estimate of unseen data. This phenomenon, which we denote *extrapolation error*, has important implications for batch and off-policy reinforcement learning, as it is generally implausible to have complete state-action coverage in any practical setting. We derive batch-constrained reinforcement learning–acting on-policy with respect to the available data, as an answer to extrapolation error. When extended to a deep reinforcement learning setting, our algorithm, Batch-Constrained deep Q-learning (BCQ), is capable of learning from arbitrary batch data, without assumptions on the collection procedure.

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

## A  MISSING PROOFS

**Lemma 1 (Batch Bellman Policy Evaluation).** *For a given batch $\mathcal{B}$, MDP $M$, and batch-constrained policy $\pi \in \Pi_{\mathcal{B}}$, the batch Bellman operator $\mathcal{T}_{\mathcal{B}}^{\pi}$ repeatedly applied to an initial $Q_0 : \mathcal{S} \times \mathcal{A} \rightarrow \mathbb{R}$, where $Q_{k+1} = \mathcal{T}_{\mathcal{B}}^{\pi} Q_k$ converges to a fixed point $Q_{\mathcal{B}}^{\pi}(s, a)$ where $Q_{\mathcal{B}}^{\pi}(s, a) = Q^{\pi}(s, a)$ for $(s, a) \in \mathcal{B}$, as $k \rightarrow \infty$.*

*Proof.* Define a new MDP $M_{\mathcal{B}}$ by restricting the full MDP $M$ to a sub-MDP with a set of states and actions with respect to $\mathcal{B}$, then the lemma follows naturally from the contraction proof of the Bellman operator (Bertsekas & Tsitsiklis, 1996).

**Theorem 1 (Batch Bellman Necessary and Sufficient Condition).** *Let $Q_{\mathcal{B}}^{\pi}(s, a)$ be the fixed point of the batch Bellman operator $\mathcal{T}_{\mathcal{B}}^{\pi}$, then $Q_{\mathcal{B}}^{\pi}(s, a) = Q^{\pi}(s, a)$ for all $(s, a) \in \mathcal{B}$ if and only if $\pi \in \Pi_{\mathcal{B}}$.*

*Proof.* ($\Rightarrow$) Follows from the definition of $\mathcal{T}_{\mathcal{B}}^{\pi}$, as if $\pi \notin \Pi_{\mathcal{B}}$ then for some $s$ we have $(s, \pi(s)) \notin \mathcal{B}$ and it follows that $Q_{\mathcal{B}}^{\pi}(s, \pi(s)) = \infty$. ($\Leftarrow$) Follows directly from Lemma 1.

**Lemma 2 (Batch-Constrained Policy Improvement).** *Let $\pi \in \Pi_{\mathcal{B}}$ be any deterministic batch-constrained policy and $\pi'(s) = \operatorname{argmax}_{a \text{ s.t.} (s,a) \in \mathcal{B}} Q^{\pi}(s, a)$, then for all $s \in \mathcal{B}$, $Q^{\pi'}(s, \pi'(s)) \geq Q^{\pi}(s, \pi(s))$.*

*Proof.* Follows from policy improvement theorem (Sutton & Barto, 1998) by restricting the MDP to a sub-MDP with the set of states and actions contained in $\mathcal{B}$ and noting $Q^{\pi}(s, \pi_{\mathcal{B}}^{*}(s)) \geq Q^{\pi}(s, \pi(s))$ for all $s \in \mathcal{B}$.

**Theorem 2 (Batch-Constrained Policy Iteration).** *Given a batch $\mathcal{B}$ and MDP $M$, then the repeated application of policy evaluation and batch-constrained policy improvement converges to a policy $\pi_{\mathcal{B}}^{*}$, such that $Q^{\pi_{\mathcal{B}}^{*}}(s, \pi_{\mathcal{B}}^{*}(s)) \geq Q^{\pi_{\mathcal{B}}}(s, \pi_{\mathcal{B}}(s))$ for all $\pi_{\mathcal{B}} \in \Pi_{\mathcal{B}}$ and $s \in \mathcal{B}$.*

*Proof.* Denote $\pi_i$ the policy at iteration $i$. By Lemma 2, we have the monotonically increasing sequence $Q^{\pi_{i+1}} \geq Q^{\pi_i}$. Noting that $Q$ is bounded above as $r$ is bounded, then policy iteration must converge to some maximum $Q^{\pi'}$, with corresponding policy $\pi' \in \Pi_{\mathcal{B}}$, such that $\pi'(s) = \operatorname{argmax}_{a \text{ s.t.} (s,a) \in \mathcal{B}} Q^{\pi'}(s, a)$ for all $s \in \mathcal{B}$. It follows that for all $\pi \in \Pi_{\mathcal{B}}$ we have $\mathcal{T}_{\mathcal{B}}^{\pi} Q^{\pi'} \leq Q^{\pi'}$, otherwise contradicting the definition of $\pi'$. Then from a repeated application of Lemma 1 we have that $Q^{\pi} \leq Q^{\pi'}$, and $\pi'$ must be the optimal batch-constrained policy.

**Corollary 1 (Batch-Constrained Optimality).** *Given a batch $\mathcal{B}$, and MDP $M$ if for all $s \in \mathcal{S}$, $(s, a^*, r, s') \in B$ where $a^* = \operatorname{argmax}_a Q^*(s, a)$ and $Q^*(s, a) = \max_{\pi} Q^{\pi}(s, a)$, then batch-constrained policy iteration converges to a policy $\pi^* = \operatorname{argmax}_a Q^*(s, a)$ for all $s \in \mathcal{S}$.*

*Proof.* Follows directly from Theorem 1 as $\pi^* \in \Pi_{\mathcal{B}}$.

**Corollary 2 (Batch-Constrained Imitation).** *Given a deterministic MDP, and trajectory $T = (s_0, a_0^*, r_0, ..., s_{i+1}, a_{i+1}^*, r_{i+1}, ...)$ where $a_i^* = \operatorname{argmax}_a Q^*(s_i, a)$ and $Q^*(s, a) = \max_{\pi} Q^{\pi}(s, a)$, then batch-constrained policy iteration converges to $\pi^*(s_i) = \operatorname{argmax}_a Q^*(s_i, a)$ for all $s_i \in T$.*

*Proof.* Follows directly from Theorem 1 by setting $\mathcal{B} = T$.

## B  RANDOM BEHAVIORAL POLICY STUDY

The experiments in Section 5 use a learned, or partially learned, behavioral policy for data collection. This is a necessary requirement for learning meaningful behavior, as a random policy generally fails to provide sufficient coverage over the state space. However, in simple toy problems, such as the pendulum swing-up task and the reaching task with a two-joint arm from OpenAI gym (Brockman et al., 2016), a random policy can sufficiently explore the environment, enabling us to examine the properties of algorithms with entirely non-expert data.

In Figure 5, we examine the performance of our algorithm, BCQ, as well as DDPG (Lillicrap et al., 2015), these two toy problems, when learning off-policy from a small batch of 5000 time steps, col-

lected entirely by a random policy. We find that both BCQ and DDPG are able to learn successfully in this off-policy task. These results suggests BCQ is less restrictive than imitation learning algorithms, which require expert data to learn. We also find that unlike previous environments, given the small scale of the environments, the random policy is able to provide sufficient coverage of the action space for DDPG to learn successfully.

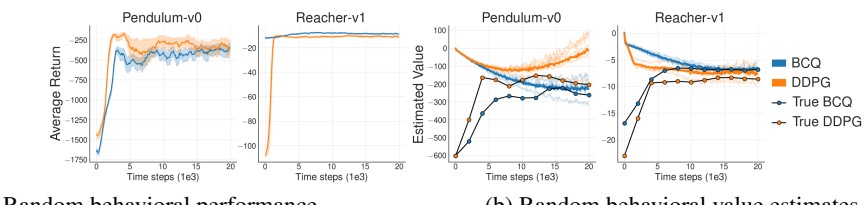

        (a) Random behavioral performance        (b) Random behavioral value estimates

Figure 5: We evaluate BCQ and DDPG on a batch collected by a random behavioral policy. The shaded area represents half a standard deviation. Value estimates include a plot of each trial, with the mean in bold. An estimate of the true value of BCQ and DDPG, evaluated by Monte Carlo returns, is marked by a dotted line. While both BCQ and DDPG perform well, BCQ exhibits more stability in the value function for the Pendulum task, and outperforms DDPG in the Reacher task. These tasks demonstrate examples where any imitation learning would fail as the data collection procedure is entirely random.

## C    EXPERIMENTAL DETAILS

Each environment is run for 1 million time steps, unless stated otherwise, with evaluations every 5000 time steps, where an evaluation measures the average reward from 10 episodes with no exploration noise. Our results are reported over 5 random seeds of the behavioral policy, OpenAI Gym simulator and network initialization. Value estimates are averaged over mini-batches of 100 and sampled every 2500 iterations. The *true* value is estimated by sampling 100 state-action pairs from the buffer replay and computing the discounted return by running the episode until completion while following the current policy.

Each agent is trained after episode by applying one training iteration per each time step in the episode. The agent is trained with transition tuples $(s, a, r, s')$ sampled from an experience replay that is defined by each experiment. We define four possible experiments. Unless stated otherwise, default settings as defined in Supplementary Material D are used.

**Batch 1 (Fixed buffer).** We train a DDPG (Lillicrap et al., 2015) agent for 1 million time steps, adding large amounts of Gaussian noise ($\mathcal{N}(0, 0.5)$) to induce exploration, and store all experienced transitions in a buffer replay. This training procedure creates a buffer replay with a diverse set of states and actions. A second, randomly initialized agent is trained using the 1 million stored transitions.

**Batch 2 (Concurrent learning).** We simultaneously train two agents for 1 million time steps, the first DDPG agent, performs data collection and each transition is stored in a buffer replay which both agents learn from. This means the behavioral agent learns from the standard training regime for most off-policy deep reinforcement learning algorithms, and the second agent is learning off-policy, as the data is collected without direct relationship to its current policy. Batch 2 differs from Batch 1 as the agents are trained with the same version of the buffer, while in Batch 1 the agent learns from the final buffer replay after the behavioral agent has finished training.

**Batch 3 (Imitation).** A DDPG agent is trained for 1 million time steps. The trained agent then acts as an expert policy, and is used to collect a dataset of 1 million transitions. This dataset is used to train a second, randomly initialized agent. In particular, we train DDPG across 15 seeds, and select the 5 top performing seeds as the expert policies.

**Batch 4 (Robust Imitation).**    The expert policies from Batch 3 are used to collect a dataset of 100k transitions, while selecting actions randomly with probability 0.3 and adding Gaussian noise $\mathcal{N}(0, 0.3)$ to the remaining actions. This dataset is used to train a second, randomly initialized agent.

## D    IMPLEMENTATION DETAILS

Across all methods and experiments, for fair comparison, each network generally uses the same hyper-parameters and architectures, which are defined in Table 1 and Figure 6 respectively. Critics and value functions follow the standard practice (Mnih et al., 2015) in which the Bellman update differs for terminal transitions. When the episode ends by reaching some terminal state, the value is set to 0 in the learning target $y$:

$$y = \begin{cases} r & \text{if terminal } s' \\ r + \gamma Q_{\theta'}(s', \pi_{\phi'}(s')) & \text{else} \end{cases} \tag{12}$$

Where the termination signal from time-limited environments is ignored, thus we only consider a state $s_t$ terminal if $t < $ `max horizon`.

Table 1: Default Hyper-parameters

| Hyper-parameter | Value |
|---|---|
| Optimizer | Adam |
| Learning Rate | $10^{-3}$ |
| Batch Size | 100 |
| Normalized Observations | False |
| Gradient Clipping | False |
| Discount Factor | 0.99 |
| Target Update Rate ($\tau$) | 0.005 |
| Exploration Policy | $\mathcal{N}(0, 0.1)$ |

```
(input dimension, 400)
ReLU
(400, 300)
RelU
(300, output dimension)
```

Figure 6: Default Network Architecture. All actor networks are followed by a `tanh · max action size`

**BCQ.** BCQ uses four networks: a perturbation model $\xi_\phi(s, a)$, a critic $Q_\theta(s, a)$, a value network $V_\psi(s)$, and a state-conditioned VAE $G_\omega(s)$, along with a target value network $V_{\psi'}(s)$. Each network in BCQ follows the default architecture (Figure 6) and the default hyper-parameters (Table 1). For $\xi_\phi(s, a, \Phi)$, the constraint $\Phi$ is implemented through a tanh activation multiplied by $I \cdot \Phi$ following the final layer.

The VAE $G_\omega$ is defined by two networks, an encoder $E_{\omega_1}(s, a)$ and decoder $D_{\omega_2}(s, z)$, where $\omega = \{\omega_1, \omega_2\}$. The encoder takes a state-action pair and outputs the mean $\mu$ and standard deviation $\sigma$ of a Gaussian distribution $\mathcal{N}(\mu, \sigma)$. The state $s$, along with a latent vector $z$ is sampled from the Gaussian, is passed to the decoder $D_{\omega_2}(s, z)$ which outputs an action. The VAE is trained with respect to the mean squared error of the reconstruction along with a KL regularization term:

$$\mathcal{L}_{\text{reconstruction}} = \sum_{(s,a) \in \mathcal{B}} (D_{\omega_2}(s, z) - a)^2, \quad z = \mu + \sigma \cdot \epsilon, \quad \epsilon \sim \mathcal{N}(0, 1), \tag{13}$$

$$\mathcal{L}_{\text{KL}} = D_{\text{KL}}(\mathcal{N}(\mu, \sigma) || \mathcal{N}(0, 1)), \tag{14}$$

$$\mathcal{L}_{\text{VAE}} = \mathcal{L}_{\text{reconstruction}} + \lambda \mathcal{L}_{\text{KL}}. \tag{15}$$

Noting the Gaussian form of both distributions, the KL divergence term can be simplified (Kingma & Welling, 2013):

$$D_{\text{KL}}(\mathcal{N}(\mu, \sigma) || \mathcal{N}(0, 1)) = -\frac{1}{2} \sum_{j=1}^{J} (1 + \log(\sigma_j^2) - \mu_j^2 - \sigma_j^2), \tag{16}$$

where $J$ denotes the dimensionality of $z$. For each experiment, $J$ is set to twice the dimensionality of the action space. The KL divergence term in $\mathcal{L}_{\text{VAE}}$ is normalized across experiments by setting $\lambda = \frac{1}{J}$. During inference with the VAE, the latent vector $z$ is clipped to a range of $[-0.5, 0.5]$ to limit generalization beyond previously seen actions. For the small scale experiments in Supplementary Material B, $L_2$ regularization with weight $10^{-3}$ was used for the VAE to compensate for the small number of samples. The other networks remain unchanged.

In the value network update, Equation (11), the VAE is sampled multiple times and passed to both the actor and critic. For each state in the mini-batch the VAE is sampled $n = 10$ times. This can be implemented efficiently by passing a latent vector with batch size $10 \cdot$ `batch size`, effectively 1000, to the VAE and treating the output as a new mini-batch for the actor and critic. When running the agent in the environment, we sample from the VAE 100 times.

**DDPG.** Our DDPG implementation deviates from some of the default architecture and hyper-parameters to mimic the original implementation more closely (Lillicrap et al., 2015). In particular, the action is only passed to the critic network at the second layer, (Figure 7), the critic uses $L_2$ regularization with weight $10^{-2}$, and the actor uses a reduced learning rate of $10^{-4}$.

```
(state dimension, 400)
ReLU
(400 + action dimension, 300)
RelU
(300, 1)
```

Figure 7: DDPG Critic Network Architecture.

As done in Fujimoto et al. (2018), our DDPG agent randomly selects actions for the first 10k time steps for HalfCheetah-v1, and 1k time steps for Hopper-v1 and Walker2d-v1. This was found to improve performance and reduce the likelihood of local minima in the policy during early iterations.

**DQN.** Given the high dimensional nature of the action space of the experimental environments, our DQN implementation selects actions over an independently discretized action space. Each action dimension is discretized separately into 10 possible actions, giving $10J$ possible actions, where $J$ is the dimensionality of the action-space. A given state-action pair $(s, a)$ then corresponds to a set of state-sub-action pairs $(s, a_i)$ for $i = \{1, ..., J\}$. In each DQN update, all state-sub-action pairs $(s, a_i)$ are updated with respect to the average value of the target state-sub-action pairs $(s', a'_j)$. The learning update of the discretized DQN is as follows:

$$y = r + \gamma \frac{1}{n} \sum_{j=1}^{J} Q_{\theta'}(s', a'_j), \tag{17}$$

$$\theta = \operatorname*{argmin}_{\tilde{\theta}} \sum_{(s,a,r,s') \in \mathcal{B}} \sum_{i} (y - Q_{\tilde{\theta}}(s, a_i))^2. \tag{18}$$

For clarity we provide the exact DQN network architecture in Figure 8.

```
(state dimension, 400)
ReLU
(400, 300)
RelU
(300, 10 action dimension)
```

Figure 8: DQN Network Architecture.

**Behavioral Cloning.** We use two behavioral cloning methods, VAE-BC and BC. VAE-BC is implemented and trained exactly as $G_\omega(s)$ defined for BCQ. BC uses a feed-forward network with the default architecture and hyper-parameters, and trained with a mean-squared error reconstruction loss.

# E  VARIATIONAL AUTO-ENCODER BACKGROUND

A variational auto-encoder (VAE) (Kingma & Welling, 2013) is a generative model which aims to maximize the marginal log-likelihood $\log p(X) = \sum_{i=1}^{N} \log p(x_i)$ where $X = \{x_1, ..., x_N\}$, the dataset. While computing the marginal likelihood is intractable in nature, we can instead optimize the variational lower-bound:

$$\log p(X) \geq \mathbb{E}_{q(X|z)}[\log p(X|z)] + D_{\text{KL}}(q(z|X)||p(z)), \qquad (19)$$

where $p(z)$ is chosen a prior, generally the multivariate normal distribution $\mathcal{N}(0, I)$. We define the posterior $q(z|X) = \mathcal{N}(z|\mu(X), \sigma^2(X)I)$ as the encoder and $p(X|z)$ as the decoder. Simply put, this means a VAE is an auto-encoder, where the a given sample $x$ is passed through the encoder to produce a random latent vector $z$, which is given to the decoder to reconstruct the original sample $x$. The VAE is trained on a reconstruction loss, and a KL-divergence term according to the distribution of the latent vectors. To perform gradient descent on the variational lower bound we can use the re-parametrization trick (Kingma & Welling, 2013; Rezende et al., 2014):

$$\mathbb{E}_{z \sim \mathcal{N}(\mu, \sigma)}[f(z)] = \mathbb{E}_{\epsilon \sim \mathcal{N}(0, I)}[f(\mu + \sigma \epsilon)]. \qquad (20)$$

This formulation allows for back-propagation through stochastic nodes, by noting $\mu$ and $\sigma$ can be represented by deterministic functions. During inference, random values of $z$ are sampled from the multivariate normal and passed through the decoder to produce samples $x$.

