# OpenReview forum: "Where Off-Policy Deep Reinforcement Learning Fails"
_ICLR.cc/2019/Conference_

### Official Review · AnonReviewer3 · 2018-11-02
**How well and why it works at states that are far from any states in the batch?**

**Rating:** 5
**Confidence:** 3

**Review:**

This paper studies extrapolation error in off-policy batch reinforcement learning (RL), where the extrapolation error refers to the overestimation of the value for the state-action pairs that are not in the training data.

The authors propose batch-constrained RL, where the policy is optimize under the constraint that, at each state, only those actions that have been taken in that state in the training data are allowed.  This is then extended to continuous space, where it allows only the state-action pairs that are close to a state-action pair in the training data.  When there is no such action for a given state, the action that is closet to a feasible action at that state is selected.

It makes intuitive sense that the proposed approach works well as long as we only encounter state-action pairs that are closed to one of the state-action pairs in the batch.  However, I do not expect that this is always the case.  The proposed method is to simply choose the closest action in the batch.  Then why does the proposed approach perform well?  Is it because the experiments are performed under rather deterministic settings?  How often are no state-action pairs found in the neighbor?  Is there any mechanism for recovering from "not in the batch"?

The paper would be much stronger if it study this challenge of "not in the batch" more in depth.  Technical contributions in the present paper are rather limited.

A key assumption in the discrete case is that whole episodes are in the batch.  This is rather restricting, because in many applications, it is infeasible to collect a whole episode, and parts of many episodes are collected from many agents.  Although this assumption is stated, it would be nice to emphasize by also stating that the theorems do not hold when this assumption does not hold.  The assumption becomes less important for continuous case, because of approximation.  It might be interesting to study the performance of the proposed approach when the assumption does not hold in the continuous case.

---

> ### Author Response · Authors · 2018-11-16
> **Response to Reviewer 3**
>
> We would like to thank the reviewer for their time, feedback and thoughts. A concern presented by the reviewer was limited technical contribution. We would like to re-emphasize our contribution towards the introduction and analysis of extrapolation error in off-policy learning. Our paper provides important insight into the working of deep reinforcement learning with finite amounts of data, or the purely “exploitative” setting, as well as imitation learning with noisy demonstrations.
>
> > It makes intuitive sense that the proposed approach works well as long as we only encounter state-action pairs that are closed to one of the state-action pairs in the batch.  However, I do not expect that this is always the case.  The proposed method is to simply choose the closest action in the batch.  Then why does the proposed approach perform well?
>
> In regions with no data at all, there is no possible mechanism for recovery because the agent, and any possible agent, will not have trained in this region. Taking the action of the closest state-action pair is an oversimplification of our method, which likely stems from the lack of clarity in our original version of Section 4.2, which has been re-written. Our BCQ algorithm produces deep network policies that can be evaluated across the entire state space and considers both the similarity of the action to the batch as well as the expected value of the action.
>
> That being said, it is important to take actions which are “close” in the Bellman update to minimize the extrapolation error in the value estimate. Otherwise, as shown in Section 3.2, there can be deterioration in performance even in regions of certainty. That is, a non-batch-constrained off-policy reinforcement learning algorithm may fail if exposed to any uncertain regions during training. Our algorithm performs well by reducing the error into the system. Informally, our value estimates are more accurate.
>
> In experiments where BCQ may take actions leading it to unseen states, such as in the experiments with an expert behavioral policy without exploration, we find that there is sufficient generalization to regions with less data to still perform well, while stabilizing the value function.
>
> For future work, an interesting extension of the algorithm would be to bias it towards regions of certainty, through an optimism-under-certainty heuristic, the polar-opposite to many exploration algorithms. This occurs implicitly in our algorithm as mimicking previously taken actions is more likely to lead to regions of certainty, but could be enforced more strongly.
>
> > A key assumption in the discrete case is that whole episodes are in the batch.  This is rather restricting, because in many applications, it is infeasible to collect a whole episode, and parts of many episodes are collected from many agents.
>
> The data doesn't need to be collected in episodic fashion, rather, that there is sufficient coverage. Collecting data in episodes is one way to ensure this, but not specifically required. This is a weaker assumption than assumptions necessary for standard Q-learning, as we no longer require visitation over all possible state-action pairs.

---

### Official Review · AnonReviewer2 · 2018-11-02
**Interesting ideas, but clarity must be improved**

**Rating:** 5
**Confidence:** 4

**Review:**

Authors consider a problem of off-policy reinforcement learning in a setting explicitly constrained to a fixed batch of transitions.
The argument is that popular RL methods underperform significantly in this setting because they fail to account for extrapolation error caused by inevitably sparse sampling of the possible action-state space.
To address this problem, authors introduce the notion of batch-constrained RL which studies policies and associated value functions only on the state-space covered by the available training data.
For practical applications a deep RL method is introduced which enables generalisation to the unseen states and actions by the means of function approximation.

I find the problem studied in the paper very important. It is indeed strongly connected to the idea of imitation learning which has been studied previously, but I like the explicit point from which authors see the problem.
The experimental results seem quite appealing to justify use of the proposed approach.

However, on the clarity side the paper should be improved before publication.

The interplay between action generating VAE G_w(s) and \pi is unclear to me.
First, what does it mean that G(s) is trained to minimise the distance D_A?

If G(s) is a VAE, then it is trained to minimise the corresponding variational lower bound, how is minimisation of the distance over actions is incorporated here? And what exactly is this distance?
Similarly, what does “D_S will be defined by the implicit distance induced by the function approximation” exactly mean?

Other comments / questions:

Page 6: “Theorem 1 implies with access to only
a subset of state-action pairs in the MDP, the value function… This suggests batch-constrained policies
are a necessary tool for combating extrapolation bias.”
This might be true, but it does not follow from the Theorem 1 as it only applies to the batch Bellman operator and not the standard one used in most methods.

Corollary 1 and 2: What is Q^* here?

Page 7, first sentence: should there be if A_s, e != \emptyset?

Epsion-Batch-constrained policy iteration: would the beam search actually maximize Q function? This needs to be proven or at least discussed.

I don’t see how is epsilon used in the iteration scheme.  This needs to be clarified.

Equation 11: the subscript of the max operator looks weird, should there be just a_i?

Figure 4: where is “True value” curve on the plots?

The notation \pi(s, a; \Phi) used throughout the paper is confusing and can be interpreted as a joint distribution over states and actions.

As I said, currently the paper does not appear to be easy to follow to me and even if it does contain important ideas, I believe they must be communicated in a clearer way.
I am eager to revise my evaluation if authors make substantial effort to improve the paper.

---

> ### Author Response · Authors · 2018-11-16
> **Response to Reviewer 2**
>
> We would like to thank the reviewer for thorough review and constructive feedback. The issues with clarity largely stemmed from Section 4.2, which we agree with the reviewer was not as clear as it could be. This section has been re-written and will hopefully satisfy the reviewer. We have removed superfluous details and simplified the presentation of Section 4.2. We believe these changes better streamlines the introduction of the (unchanged) algorithm, and better justifies some of the algorithmic choices. Other small adjustments to notation and clarity have been made throughout the paper, with regards to both your comments as well as the other reviewers.
>
> Further Responses to Comments:
>
> > Page 6: “Theorem 1 implies with access to only a subset of state-action pairs in the MDP, the value function… This suggests batch-constrained policies are a necessary tool for combating extrapolation bias.” This might be true, but it does not follow from the Theorem 1 as it only applies to the batch Bellman operator and not the standard one used in most methods.”
>
> The claim we intended to make was not that batch-constrained policies are necessary, but rather suggest that they are likely, or potentially, necessary. We have clarified this in the paper.
>
> > Figure 4: where is “True value” curve on the plots?
>
> Initially we left out the true value curve to allow for a larger figure, putting more emphasis on the results. We have re-added the true value curve.

---

### Official Review · AnonReviewer1 · 2018-11-05
**Solid approach to applying RL algorithms to batch imitation learning from noisy demonstrations**

**Rating:** 7
**Confidence:** 4

**Review:**

Summary:
Proposes BCRL for learning from a fixed collection of off-policy experience (I'll call this the "training data"). BCRL attempts to avoid backing up values from states that are not present in the training data, on the assumption that the current estimates of these values are likely to be inaccurate. In the continuous state-action case, this is accomplished by training a generative model to propose, given a state `s`, an action `a` such that a transition similar to `(s, a)` is in the training data. A secondary policy is then trained to perturb the proposed action within a constrained region to maximize value. BCRL outperforms DDPG and DQN when learning from fixed data, but BCRL is slightly worse than behavior cloning at learning to reproduce an expert policy that does not take exploration actions.

Review:
The overall approach is sound. The problem of extrapolation is intuitively obvious, but not something I had thought about before. I think typically exploration would correct the problem since states with over-estimated values would become more likely to be reached, giving an opportunity to get a better estimate.

The learning setting is closer to imitation learning than to what I would call RL, since the BCRL approach essentially avoids extrapolation error by ignoring the parts of the problem that are not represented in the training data. The well-known problem with behavior cloning is compounding errors once the agent strays into areas of the state space that are far from the training data. To me "off-policy RL" implies that the goal is to learn a complete policy from off-policy data. I think the "competitors" to which BCRL should be compared are imitation learning algorithms address noisy demonstrations, and not so much off-policy RL algorithms. It would also be interesting to see the generalization performance of BCRL outside of its training data.

The BCRL idea might be applicable in a conventional RL setting as well, since the initial stages of learning could be subject to a similar extrapolation error until there has been enough exploration. A comparison to something like TRPO in this setting would be interesting.

The paper is well-written with good coverage of related literature. There are a few points where the technical content is imprecise, which I note below.

Comments / Questions:
* Could one obtain a similar effect to BCRL by simply initializing the value estimates pessimistically?
* Sec 4.1: Since B is a set of (s, a, s', r) tuples, what does it mean for a state s' to be "in B"? Similar question for state-action tuples (s, a).
* As you note in the appendix, the construction in Sec 4.1 is essentially creating a new MDP that contains only the transitions that occur in the training data. I'd suggest stating as much in the main paper for intuition.
* Sec 4.2 / 5: The perturbation constraint \Phi is set to 0.05 in the experiments. Since the actions in these control problems are vectors, what does a scalar constraint correspond to? How is the constraint enforced during learning?
* What are the distance functions D_S and D_A?

Pros:
* A good approach to applying RL methods in the "imitation-like" setting. I've seen similar things attempted before, but this method makes more sense.

Cons:
* The learning setting is more like "fuzzy" behavior cloning from noisy data than off-policy RL. Experimental comparison against more-sophisticated imitation learning approaches is missing.

---

> ### Author Response · Authors · 2018-11-16
> **Response to Reviewer 1**
>
> We would like to thank the reviewer for their helpful comments and positive feedback. We have added an experimental result to the supplementary material to distinguish ourselves further from imitation learning algorithms and made several clarifying statements and adjustments based on your recommendations.
>
> One con listed was missing comparisons against other state of the art imitation learning algorithms which are robust to noisy demonstrations. However, to the best of our knowledge, we are not aware of any which satisfy the batch setting, where no further data is collected, while also setting no requirements on data being labelled expert vs. non-expert. One algorithm which does satisfy these conditions is [1], but only operates in with discrete actions, making it weak baseline in a continuous control benchmark, where independent discretization would be required. If there was a particular algorithm you had in mind when writing the review, we would be happy to include it in the final paper.
>
> We also note the line between off-policy and robust imitation is fairly thin. For example, in the tabular setting, our approach can learn from the set of data that includes all state-action pairs, similarly to off-policy learning. All state-action pairs, of course, also includes expert actions and could be considered a robust imitation learning algorithm as well. An expert behavioral policy is necessary for the data collection process to be sufficiently interesting, as a purely randomly policy doesn’t cover enough of the state space for it to be possible to learn meaningful behavior. To further demonstrate the effectiveness of our algorithm as an off-policy algorithm, we included results with a purely random behavioral policy on a pendulum and reacher task in the supplementary material B, where the state space can be sufficiently covered by taking random actions.
>
> Further Responses to Questions/Comments:
>
> > Could one obtain a similar effect to BCRL by simply initializing the value estimates pessimistically?
>
> Essentially yes, especially in the tabular setting, however, this would slow learning as it may take many updates to “wash away” initial negative bias. Furthermore, in a function approximation setting, maintaining an optimistic or pessimistic initialization over many timesteps is impractical and often implausible. Finally, for a fixed, non-batch-constrained policy, this also gives biased estimates. Introducing the notion of batch-constrained gives some understanding to when the policy would be biased vs. when it wouldn't.
>
> > Sec 4.1: Since B is a set of (s, a, s', r) tuples, what does it mean for a state s' to be "in B"? Similar question for state-action tuples (s, a).
>
> s' in B is shorthand for (s, a, s', r) in B for some s, a, r. We have added a clarifying sentence in the background.
>
> > As you note in the appendix, the construction in Sec 4.1 is essentially creating a new MDP that contains only the transitions that occur in the training data. I'd suggest stating as much in the main paper for intuition.”
>
> At your recommendation we have added this to the main paper.
>
> > * Sec 4.2 / 5: The perturbation constraint \Phi is set to 0.05 in the experiments. Since the actions in these control problems are vectors, what does a scalar constraint correspond to? How is the constraint enforced during learning?
>
> This correspond to \Phi * I * tanh() following the final layer. We have added a clarifying sentence in the supplementary.
>
> References:
> [1] Gao, Yang, Ji Lin, Fisher Yu, Sergey Levine, and Trevor Darrell. "Reinforcement learning from imperfect demonstrations." arXiv preprint arXiv:1802.05313 (2018).

---

### Public Comment · ~David_Schneider1 · 2018-10-09
**question on example of extrapolation error in simple example**

I'm trying to understand equation (6) in section 3.1. Why is Q(·, a1) = 1 + γQ(s1, a0), that is for both Q(s0, a1) and Q(s1,a1), we have the same value? Doesn't equation (4)  given the sample (s0,a1,r=1,s1) in the batch, define

Q(s0, a1) = k(s0,s0) * (1 + γ V(s1))

and

Q(s1, a1) = k(s1,s0) * (1 + γ V(s1))

so that is would depend on the kernel function k? It seems a natural kernel might be one where k(s0,s0)=1 but k(s1,s0) ~ 0.

---

> ### Author Response · Authors · 2018-10-09
> **Re: question on example of extrapolation error in simple example**
>
> Hi, thanks for your question. Indeed, the value is usually dependent on the kernel function, but this weighting is normalized over all examples of the corresponding action (equation 5). With only one example of a1 the kernel term will be reduced to 1.

---

> > ### Public Comment · ~David_Schneider1 · 2018-10-09
> > **Thanks!**
> >
> > I follow now

---

### Public Comment · ~David_Schneider1 · 2018-10-09
**What about MDP's where the batch doesn't show all the next states you get to after an action?**

I really appreciate this paper. It clearly explains of a lot of issues with off-policy I've been realizing recently.

I think there is an issue with Theorem 1 and the proof of Lemma 1 - in my mind, it is fixed by some assumption about the environment.  The issue I see is that the batch of data, for the state/action pairs we get to observe in the batch, may mis-represent the underlying MDP restricted to those states.

For instance, suppose there is a (s,a) pair where the action sometimes leads to 'success' and sometimes 'failure' - the terminal state. Like the state 's' could be the robot faces a pit, and the action 'a' could be a jump to get over it. Most of the time perhaps, the action leads to failure - the terminal state, but we were unlucky in our batch and we saw success.

If you assume they next state is deterministic based on the action - which may be reasonable for continuous robot control - where there are some precise values that always lead to success with the jump - I think you could get around this, but if not - like in Theorem 1 - I don't see how  Q^{\pi}_{B}(s,a) = Q^{\pi}(s,a), the first is the Q-value we learn from the batch, where we never see a failure after (s,a) so our Q will be high- but the latter is the true Q-value over the whole MDP - where we often see failure, and the Q would be low.

Likewise in the proof of Lemma 1 - I think there is one natural new MDP M_{B} to define based on the data you observed, but you need to estimate the transitions from that data - and you might miss things. Like if you see

(state, action, next_state):
(s0, a0, s1)
(s0, a0, s1)
(s0, a0, s2)

you'd naturally set the transition P(s1 | s0, a0) = 2/3 and P(s2 | s0,  a0) = 1/3 -  but this might not be the true transitions. The fixed point you converge to will depend on the MDP you derive from your observations.

---

> ### Author Response · Authors · 2018-10-09
> **Re: What about MDP's where the batch doesn't show all the next states you get to after an action?**
>
> Thank you for your interest in our paper! From what is stated at a theoretical level, Lemma 1 and Theorem 1 are not broken by non-determinism.
>
> The Bellman operator assumes access to the underlying MDP as it includes an expectation over the next state. The batch Bellman operator, which we introduce, is an extension which masks out any unseen state-action pairs by setting their value to infinity. As a result, the batch Bellman operator can still access the true expectation, even with only a sample of the possible transitions in the batch, allowing Theorem 1 and Lemma 1 to hold.
>
> That being said, I believe the point you are making has to do with the more practical scenario, say with a batch-constrained tabular Q-learning-- what happens if we only have a sample of the possible transitions, without access to the true expectation? Our approach makes the same practical assumption as other off-policy algorithms such as Q-learning and KBRL, which is that the samples you do have for (s,a) are representative of the true MDP. In this case, as you have stated, there is no guaranteed convergence to the true Q^pi(s,a), for any off-policy algorithm, unless the samples you have are indeed representative, e.g. the environment is deterministic or there is infinite data. And of course, we demonstrate the effectiveness of our approach in more complex settings in Section 5.

---

### Author Response · Authors · 2018-11-16
**General Response and Overview**

We would like to take this opportunity to thank each reviewer again. We found that the quality of the reviews was high and the reviewers made insightful commentary, each with different flavors with respect to the paper. As this paper introduces the first analysis into the batch setting with deep function approximation, it makes sense that there would be small issues in clarity, and as displayed by our lengthy related works, there are many possible interpretations to where, and how, our work is insightful or significant. We have carefully responded to each reviewer and have made many small updates throughout the paper to improve the clarity and dispel any confusion about the task as well as our approach.

As mentioned in our response to Reviewer 2, almost all significant issues with clarity were from Section 4.2 which we have re-written to better justify the design choices made when approximating the batch-constraint in a continuous setting. We have expanded the introduction on extrapolation error to be more explicit on its origin. Additionally, although no reviewer took issue with the original title, we have taken the opportunity to modify the title to be more informative towards the contents of the paper. We believe these changes address the reviewers’ concerns and greatly improve the quality of the paper. We will continue to edit the paper before the deadline and our happy to respond to further comments, questions or concerns.

---

### Public Comment · (anonymous) · 2018-11-20
**Questions on implementation details**

I really enjoyed reading this paper! I had a few questions about some implementation details (I wanted to give the algorithm a try myself), and I was hoping that you could help me out.

In the algorithm description (Algorithm 1 in the paper), could you explain how the perturbation model is updated in more detail. Equation 10 is equally confusing. I'm trying to understand whether the actions should come from a sampled mini-batch or if they should come from the VAE decoder when (1) passing them to the perturbation model to compute perturbations and (2) when actually perturbing the actions before passing them to the critic network.

Could you also give me some more intuition on why a perturbation model is trained? It seems as though we could just leave the perturbation model out and just do Q-learning using the VAE, critic, and value networks.

Finally, equation (11) implies that the value function is trained on a mini-batch of next states instead of current states. Is there a reason why the value function loss isn't just over the current states (s) instead of the next states (s')?

Also I wanted to point out a few typos (although I could be mistaken):

-Equation (8) should have a (+) sign for the KL term.

-Equation (16) should have a factor of 0.5 outside the sum and it should be the log variance, not the log standard deviation.

-The line in Algorithm 1 right before the "Update VAE" line should have the VAE model take the mini-batch of actions as input along with the mini-batch of states.

---

> ### Author Response · Authors · 2018-11-20
> **Re: Questions on implementation details**
>
> Hi, thanks for your interest in the paper, and for catching a few typos! When the paper is de-anonymized we will include a GitHub repository in the paper. Until then, we have included the core algorithm code in an anonymous pastebin ( https://pastebin.com/UTaiR5ZZ ), which should speed up your implementation.
>
> > In the algorithm description (Algorithm 1 in the paper), could you explain how the perturbation model is updated in more detail. Equation 10 is equally confusing. I'm trying to understand whether the actions should come from a sampled mini-batch or if they should come from the VAE decoder when (1) passing them to the perturbation model to compute perturbations and (2) when actually perturbing the actions before passing them to the critic network.
>
> The actions can come from anywhere. Either the mini-batch, randomly generated, or from the VAE. During the target update and during test time, the perturbations are applied to actions generated by the VAE, so it makes the most sense to train them on the same distribution.
>
> Equation 10 describes the deterministic policy gradient algorithm (DPG). The perturbation model is updated so that the perturbed actions maximize the critic. Normally with DPG, there is a policy pi(s) which is trained to maximize Q(s,pi(s)). In our method, instead of pi(s) which can output any action, we have xi(s,a) + a, where xi(s,a) can only output in a limited range.
>
> > Could you also give me some more intuition on why a perturbation model is trained? It seems as though we could just leave the perturbation model out and just do Q-learning using the VAE, critic, and value networks.
>
> You could, but as discussed in the paper, using a perturbation model gives an increase in flexibility in the action selection. For example, if the VAE outputs 10 similar actions, allowing perturbations gives more diversity in actions that could be possibly selected. Another case is where the entire action space has been adequately covered, and any action could be viably generated by the VAE. To see the highest valued action would likely require sampling from the VAE many times. Allowing perturbations avoids this issue.
>
> > Finally, equation (11) implies that the value function is trained on a mini-batch of next states instead of current states. Is there a reason why the value function loss isn't just over the current states (s) instead of the next states (s')?
>
> In principle it doesn't matter, but the code does in fact train over s, as shown in Algorithm 1. Thanks for catching this, along with the other typos. We have uploaded a corrected version.

---

> > ### Public Comment · (anonymous) · 2018-11-21
> > **Minor Clarification**
> >
> > Thanks a lot for posting the code snippet and the extremely detailed answer - I really appreciate and found it incredibly helpful!
> >
> > I had one minor follow-up question. When computing the VAE loss, in the code sample you provided you simply added the reconstruction loss with the KL loss, but in the paper, you mentioned that the weighting of the KL term is inversely proportional to the size of the latent dimension. Also, it seems like the latent vectors are only clipped when sampling from the decoder directly (not during training). Could you please confirm? Thanks again!

---

> > > ### Author Response · Authors · 2018-11-21
> > > **Re: Minor Clarification**
> > >
> > > Hi again, glad the response was helpful!
> > >
> > > (1) The inverse weighting happens by taking the mean in the KL divergence term (line 150) rather than the sum. (2) Good catch! Yes, the clipping only occurs during inference, the VAE training itself is not modified. We have updated the supplementary to reflect this.

---

### Public Comment · (anonymous) · 2018-12-09
**Relationship with previous work?**

Off-policy Q learning is already known not to converge even with linear function approximation [1]. What is the new insight and the relationship between the example listed here, and the previous work?

Reference
[1] Residual Algorithms: Reinforcement Learning with Function Approximation

---

> ### Author Response · Authors · 2018-12-10
> **Re: Relationship with previous work?**
>
> It is true that specific/adversarial counter-examples exist for most forms of function approximation. However, in this work we examine environments where deep Q-learning algorithms have already been shown to perform well on. We show that given the same dataset, deep off-policy algorithms can perform very differently depending on how close their policy is to the policy which generated the dataset. This is an important insight because we show that off-policy algorithms, which would otherwise perform well, can now fail. Furthermore, we demonstrate how this issue can be corrected and introduce a practical algorithm.

---

### Meta-Review · Area_Chair1 · 2018-12-14
**Nice work with potential, but contributions need to be strengthened**

**Confidence:** 5
**Recommendation:** Reject

**Metareview:**

The paper proposes batch-constrained approach to batch RL, where the policy is optimized under the constrain that at a state only actions appearing in the training data are allowed.  An extension to continuous cases is given.

While the paper has some interesting idea and the problem of dealing with extrapolation in RL is important, the approach appears somewhat ad hoc and the contributions limited.

For example, the constraint is based on whether (s,a) is in B, but this condition can be quite delicate in a stochastic problem (seeing a in s *once* may still allow large extrapolation error if that only observed transition is not representative).  Section 4.1 gives some nice insights for the special finite MDP case, but those results are a little weak (requiring strong assumption that may not hold in practice) --- an example being the requirement that s' be included in data if (s,a) is in data and P(s'|s,a)>0 [beginning of section 4.1].

In contrast, there are other more robust and principled ways, such as counterfactual risk minimization (CRM) for contextual bandits (http://www.jmlr.org/papers/v16/swaminathan15a.html).  For MDPs, the Bayesian version of DQN (the cited Azizzadenesheli et al., as well as Lipton et al. at AAAI'18) can be used to constrain the learned policy as well, with a simple modification of using the CRM idea for bandits.  Would these algorithms be reasonable baselines?